# Residual Disease in Glioma Recurrence: A Dangerous Liaison with Senescence

**DOI:** 10.3390/cancers13071560

**Published:** 2021-03-29

**Authors:** Diana A. Putavet, Peter L. J. de Keizer

**Affiliations:** Center for Molecular Medicine, Division LAB, University Medical Center Utrecht, 3584CG Utrecht, The Netherlands; d.a.putavet@umcutrecht.nl

**Keywords:** GBM, senescence, SASP, residual disease, wnt, NFkB, glioma stem cells, CD44, IL8

## Abstract

**Simple Summary:**

Glioblastoma (GBM) is an aggressive and mostly incurable from of brain cancer. This is largely due to individual tumor cells invading the surrounding tissue, through which they evade surgical removal and reconstitute the tumor. Here, we define three distinct environments that GBM cells migrate into: the vascular, the neural, and the glial niches. These environments can nurture GBM recurrence through factors associated with a process called “senescence”; a cellular stress response which causes them to secrete mostly pro-tumorigenic factors. Senescence is especially relevant to late-stage brain cancer since it occurs mainly in aged people, who have senescent cells. Moreover, brain cells can become senescent in response to unresolved damage from chemo- and radio- therapies. As such, we summarize recent literature on brain senescence and discuss strategies to optimize and implement anti-senescence therapies aimed at overcoming recurrence and lethality from GBM.

**Abstract:**

With a dismally low median survival of less than two years after diagnosis, Glioblastoma (GBM) is the most lethal type of brain cancer. The standard-of-care of surgical resection, followed by DNA-damaging chemo-/radiotherapy, is often non-curative. In part, this is because individual cells close to the resection border remain alive and eventually undergo renewed proliferation. These residual, therapy-resistant cells lead to rapid recurrence, against which no effective treatment exists to date. Thus, new experimental approaches need to be developed against residual disease to prevent GBM survival and recurrence. Cellular senescence is an attractive area for the development of such new approaches. Senescence can occur in healthy cells when they are irreparably damaged. Senescent cells develop a chronic secretory phenotype that is generally considered pro-tumorigenic and pro-migratory. Age is a negative prognostic factor for GBM stage, and, with age, senescence steadily increases. Moreover, chemo-/radiotherapy can provide an additional increase in senescence close to the tumor. In light of this, we will review the importance of senescence in the tumor-supportive brain parenchyma, focusing on the invasion and growth of GBM in residual disease. We will propose a future direction on the application of anti-senescence therapies against recurrent GBM.

## 1. Residual Disease Is the Main Contributor to Treatment Failure of GBM

Gliomas are the most common type of cancer of the central nervous system, whereas the early grades I–II are self-limiting and can often be removed by resection, grade III and IV are characterized by a diffuse spread into the neighboring brain [1]. Stage IV gliomas, also known as Glioblastoma, or previously called Glioblastoma multifome (GBM), span well over half of all gliomas and some 15% of all types of brain cancer combined [1,2]. GBM prevalence is on the rise, with an incidence that doubled over the last two decades [3,4]. GBM has one of the worst prognoses of all human cancers, with as little as 4–6.7% of patients surviving to five years after diagnosis and only 0.7% making it to ten years [5,6,7]. Based on this, we will review what causes current therapies to fail and explore experimental areas of therapy development. 

Tumor cell invasion into the normal brain poses one of the primary hurdles for GBM treatment success [8] (Figure 1, Part 1). Maximally safe surgical resection usually includes an area well beyond the tumor [1] and several retrospective studies suggest that larger extents of resection delay recurrence and prolong overall survival [9,10,11,12]. Unfortunately, the depth of surgery is inevitably limited by the cerebral anatomy to avoid compromising neurological function [13]. Even with a seemingly complete resection of the tumor, infiltrated residual cells often remain behind and give rise to recurrence. In line with this, postmortem studies have shown that neoplastic cells extend beyond the visible enhancement of CT and MRI images [14,15]. This makes surgery non-curative and renders GBM a malignant disease.

Since most of the GBM invades locally, it is common practice to apply radiotherapy (60 Gy total, 2 Gy/day) to the original tumor area plus a small margin to encompass residual cells [16]. The DNA alkylating chemotherapy Temozolomide (TMZ) is administered together with radiotherapy, followed by additional adjuvant TMZ and tumor-treating fields (TTFields) [1,17]. Ultimately, tumors recur within as little as half a year from diagnosis on average (Figure 1, Part 1), and, in most patients (up to 90%), the tumor regrows within 2 cm from the resection margin [5,6,18,19]. At this stage, the recurrent disease has no specific standard-of-care. Surgery, radiotherapy, TTFields, and systemic chemotherapy, including TMZ and bevacizumab, are options (as reviewed in [20]). Still, these interventions only have modest benefits (as reviewed in [21]), and a substantial proportion of patients are not eligible for second-line therapy [22]. Thus, it is crucial to understand what causes residual GBM cells to escape cell death and migrate into the brain parenchyma to cause lethality ultimately. Here, we focus on the role of aging and provide arguments that a process that drives aging, cellular senescence, should be considered for the development of new treatment options against GBM.

## 2. Aging Is a Risk Factor for Advanced GBM and Treatment Resistance—A Connection with Senescence?

The etiology of gliomas is still largely unclear. One of the few established risk factors is radiation exposure, and a small percentage is linked to sporadic hereditary predisposition syndromes [1]. There is one other clear risk factor for the development of the more aggressive glioma, which is also a prognostic factor: Age at diagnosis. In contrast to lower grade glioma, stage IV, GBM, is primarily diagnosed in older individuals, with a median age of 62 at diagnosis for isocitrate dehydrogenase (IDH) wild type tumors (Figure 2). Age at diagnosis negatively affects survival rate [7]. To illustrate, patients who reach five years following diagnosis are mostly, approximately 20%, under the age of 50, whereas only 5% of patients between the ages of 50 and 64 and much less for the elderly over 65 years of age, 0.2% [20,23]. This poor prognosis is primarily attributed to the extent of resection and poor performance or frailty associated with advanced age [23,24]. However, this may in part be caused by brain aging. Age-related inflammation may promote GBM progression, as has been observed for other neoplasms [25]. One important pathognomonic landmark of the aging brain is senescence [26]. Not surprisingly, senescence is one of the main drivers of aging, and it also supports cancer progression. This circumstantial evidence compels an investigation into whether brain senescence has a causative role in glioma progression, as senescence could be an attractive target in light of emerging anti-senescence strategies. 

## 3. Senescence Is a Causal Driver of Aging and Cancer

Since the discovery of “cellular senescence” over half a decade ago, it has been proven beyond doubt to be a major driver of aging [28,29,30,31,32]. The senescence-associated growth arrest requires the coordinated action of the p53 and p16^INK4A^ tumor-suppressive pathways. Uncapped telomeres and DNA double-strand breaks activate a DNA damage response that leads to stabilization of p53 initiating transcription of the cyclin-dependent kinase inhibitor (CDKi) p21 [33]. After this initial arrest, a permanent arrest is controlled by p16 ^INK4A^. Senescent cells arise during wound healing responses, but the immune system quickly clears them; however, with advanced age, they evade the immune-mediated clearance and are maintained in various tissues, including the brain [28].

Although senescent cells make up a small percentage of the cells in an old organ, they can affect pathology most likely through their secretory phenotype termed the senescence-associated secretory phenotype (SASP) [34,35]. Significantly, senescence is involved in every cancer progression step (reviewed in [36,37]). Specifically, senescent cells can promote tumor cell proliferation, invasion, and metastasis in a few different types of cancer [38,39]. To understand whether a senescent brain could coordinate tumor progression, this review aims first to define invaded GBM cells in the context of their peritumoral microenvironment and second to describe how a senescent microenvironment could spur GBM progression. Since the main feature of senescence that drives cancer progression is the SASP, we will highlight critical observations for SASP components essential to growth and invasion into the peritumoral niche. These will comprise the criteria used to score the role of senescence in glioma.

## 4. The Identity of the Tumor Cells in the Peritumoral Zone

### 4.1. Heterogeneity and Plasticity at the Tumor’s Edge by Pro-Inflammatory Signals

GBM is highly heterogeneous, as implied by the previously-used appellation multiforme, not only on histology but also at the molecular level. Driver mutations differ significantly among tumor cells [40,41,42]. GBM can be classified into four subtypes based on gene expression and mutational data: classical, mesenchymal, neural, and proneural [43]. The proneural subtype has been associated with the presence of oligodendrocytic genes. Nearly all cases have point mutations in IDH1, and more than half show TP53 loss of hemizygosity and altered platelet-derived growth factor (PDGFR) expression. The neuronal subtype mostly expresses neuronal markers. Nearly all of the classical subtype presents an amplified epidermal growth factor receptor (EGFR) signaling, wild type TP53, and deletion of CDKN2A. This subtype is marked by high Notch and Sonic Hedgehog markers. The mesenchymal subtype has a higher expression of mesenchymal and astrocytic markers such as CD44. It has mutated TP53, neurofibromatosis (NF1), and Phosphatase and Tensin homolog (PTEN) genes and increased serine-threonine kinase (Akt) and NFkB pathways. Strikingly, a gene expression study in single cells shows that all four signatures identified by Verhaak et al. may be present in the same tumor [44]. Moreover, a mathematical model predicts that cells can regularly switch among them and that these subtypes are plastic cell states rather than rigid subtype [45]. Single-cell approaches using RNA sequencing and phosphoproteomics have shown that the four phenotypic signatures are dynamic and interchangeable [44,46,47]. This marked plasticity is probably a consequence of cancer cells undergoing phenotypic modifications to adapt to environmental challenges and complicate efficient treatment effectively. Among the four subtypes, patient survival is quite similar. However, the classical subtype has the best prognosis, and the mesenchymal subtype has the worst prognosis [43]. The next question is whether such marked heterogeneity and plasticity contribute to therapy resistance enabling recurrence. 

Molecular profiling of recurrent disease has revealed that both heterogeneity and plasticity are involved in therapy resistance in GBM, as is the case for other cancer types. Recurrent disease shares only half of its genetic mutations with primary tumors [45]. This implies that, in half of the tumors, it is likely that new clones arose during treatment but also that, in the rest, existing clones took over in the recurrence. Notably, more than half of recurrent GBM is of the mesenchymal subtype, suggesting that this subtype might endow chemo-resistance or at least a survival benefit [48]. This is confirmed in glioma mouse models. Tumors with a proneural phenotype switch to mesenchymal in response to ionizing radiation [49]. Mesenchymal transformation can promote resistance to classical chemotherapies in other neoplasms, and mesenchymal transition can promote radiotherapy resistance in GBM cell lines as well [45,49,50,51,52]. Ionizing radiation, a component of the standard of care for GBM, can induce mesenchymal features in GBM through NFkB activation [53,54]. NFkB may drive mesenchymal conversion by controlling the expression of genes associated with a mesenchymal program such as Signal Transducer and Activator of Transcription (STAT) 3 and CCAAT-enhancer-binding protein B (C/EBPB) [55]. NFkB is constitutively activated in glioma stem cell patient lines and GBM mouse models and has an essential role in controlling GBM pathobiology [56,57,58,59]. Thus, a mesenchymal state enhances GBM migration and therapy resistance, and can, paradoxically, be created in response to chemo-radiotherapy.

Using MRI-localized biopsies, numerous studies identified differences in gene expression and mutational signatures between the contrast-enhancing core and the non-enhancing margins of tumors [60,61]. Analysis of such biopsies revealed elevated NF-kB protein expression in invasive GBM cells at the tumor margin compared to non-invasive cells within the cellular core [62]. Moreover, cells at the invasive edge show exceptionally high plasticity [53,63]. Although NFkB inhibition did not alter GBM cell proliferation, it strongly reduced cell invasion. In contrast, Myelocytomatosis oncogene cellular homolog (Myc) inhibition decreased cell proliferation and, in turn, increased cell migration. Such evidence underscores plasticity between an MYC-driven proliferative, yet non-invasive, tumor core versus a less proliferative but invasive, NFkB-driven tumor rim. Interestingly, NFkB can be activated by hyaluronan, which is enriched at the tumor margin, through engaging the toll-like receptor (TLR) 4 on the surface of CD44+ glioma stem-like cells [64,65]. NFkB is responsible for maintaining the stem-cell identity since inhibition thereof resulted in senescence and differentiation of glioma stem-like cells [66]. This makes NFkB of relevance to induction and maintenance of a pro-inflammatory microenvironment. The question is how this is connected to GBM invasion and therapy resistance.

### 4.2. The GBM Tumor Micro-Environment (TME) Determines GBM Cancer Stemness 

Glioma stem cells (GSC) coexist with non-stem bulk cells and are likely to dynamically inter-change, stimulating the microenvironment [67]. So far, GBM does not have an unequivocally universal GSC marker. The expression of particular cell membrane antigens, including CD133 and CD44, and neural stem and progenitor cell markers, such as Nestin and Olig2, has been used to identify the GSC subsets. Although several studies point out the tumorigenic capacity of GSCs [68], Dirkse et al. provide strong evidence that GSCs do not constitute a defined cellular entity but rather a cellular state adapting to microenvironmental cues [69]. They also found that reconstitution of heterogeneity drives tumor growth in vivo, indicating that tumorigenic potential is defined by plasticity rather than CSC multipotency. In addition, CD133 shows a dynamic expression pattern, with CD133 low cells capable of upregulating CD133 transplanted in a nude mouse [70]. The niche can coordinate such plasticity. Accordingly, pro-inflammatory cytokines from the TME, such as tumor necrosis factor (TNF) alpha, interleukin(IL) 6 and 8, can activate NFkB and STAT3, helping maintain a GSC identity and instate chemo-resistance [71]. Thus, a pro-inflammatory TME is a driving force in GSC development. 

GSC show enhanced therapy resistance, thereby contributing to recurrence [72]. The first evidence supporting this came from a study conducted by Chen [73] where they used a reporter to tag Nestin + cells in a murine glioma model. Speaking to the essential role of GSC in tumor growth, ablating these Nestin + cells prevented tumor growth, and, with the addition of TMZ, it stopped tumor development altogether. Nestin + cells present at the tumor margin survived TMZ therapy, recovered from the chemotherapy-induced growth arrest, and reconstituted the tumor leading to recurrence [73,74]. TMZ and radiation enriched for cells expressing CD44 [68]. In line with this, recurrent tumors are enriched for CD44 [68]. GSCs have inherent properties that make them more resilient to therapy. CD44 + cells exhibit an enhanced basal expression of DNA damage repair (DDR) factors, which endows them with a superior detection of DNA damage and improved DDR, leading to a lower burden of double-stranded breaks from therapy [75,76,77]. In addition, CD44 is highly expressed in the mesenchymal subtype, which is enriched upon recurrence [78]. Moreover, GBM that primarily expresses CD44 at the margin compared to the tumor core has a worse prognosis than at tumors where CD44 is evenly distributed [64]. Thus, stemness, e.g., caused by chemo-radiotherapy-induced NFkB activity, is an attractive target for intervention against GBM therapy resistance leading to recurrence. Moreover, the presence of these cells in the peritumoral brain reinforced their role in residual disease. How these cells end up in the healthy brain will be discussed in the next section.

## 5. The Right Niche to Subsist in the Peritumoral Zone

### 5.1. Enhanced GBM Invasion Features

One of the main clinical hallmarks of high-grade glioma is extensive infiltration into the peritumoral healthy brain. Not surprisingly, gliomas express many factors that promote invasion (as reviewed [79]. Moreover, migratory behavior of patient-derived tumor cells from in vitro may predict clinical outcomes, including recurrence [80]. GBM infiltrates peritumorally in two different patterns: diffusely as single cells and collectively as a strand. These two behaviors were documented using intravital imaging to track the real-time movement of the mouse glioma line GL261 out of the primary tumor in mouse brains [81]. Collective invasion has been observed in patient samples as well using a three-dimensional (3D) image reconstruction of 30 to 50 conventional single-slice pathology images of human glioma samples [82]. Collective invasion is directional and can lead to faster growth of the tumor into the surrounding brain and more toxicity from disrupting vascular integrity and neurotoxicity and seizures, one of the earliest signs of GBM [81,82,83,84]. Invasion and growth into the peritumoral zone depend on the brain’s extracellular matrix (ECM), which can be either a barrier or a scaffold for invading cells.

In contrast to other organs, the brain parenchyma consists mostly of proteoglycans and their binding partners: hyaluronan (HA) and tenascins, and little fibronectin and collagens primarily expressed on blood vessels [85]. Experiments using de-cellularized brain tissues suggest that the ECM is strongly involved in single-cell infiltration [86]. The main structural component of the brain ECM, HA, stimulates migration through binding to its cognate receptor CD44 [87,88,89,90]. Gliomas are enriched for specific ECM components. For example, fibronectin, which is typically expressed in blood vessels in the healthy human brain, is also expressed in gliomas, specifically in the mesenchymal subtype [91]. It enables cohesion between glioma cells favoring collective invasion [91]. Furthermore, GBM-derived fibronectin could have an additional role in chemo-resistance since it suppressed chemotherapy-induced apoptosis through integrin signaling [91,92]. 

The interaction with the ECM is mediated through integrins on the cell surface, regulating signaling pathways that control cytoskeletal organization, critical for cell migration. Indicating their importance, tumors upregulate multiple integrins with increasing grades compared to the normal brain [93]. To further migration, ECM–cell interactions are dynamic with constant detachment and re-attachment of cells from the ECM. Thereby, glioma cell movement appears saltatory in intravital microscopy and ex vivo microscopy [81,94,95]. To enable this migration, matrix remodelers such as the metalloproteinases (MMP) 2 and 9 control the ECM’s integrity. Underscoring their importance, both are highly expressed in glioma than normal brains [96,97]. Besides degrading ECM, they both activate cytokines and chemokines associated with growth and invasion, including the IL1 and IL8 families [98]. MMP2 appears to be specifically essential to migrate through collagen type I, mostly found in the tumor core and the perivascular space [8,86]. MPPs are typically secreted by the tumor cells or the brain parenchyma. As most tumors recur within 2 cm from the resection edge, it is reasonable to hypothesize that the peritumoral niche cells could influence this invasion [19]. Amassing recent literature, we identified three distinct niches that promote both tumor growth and invasion (Figure 1, Part 2): (A) the vascular niche, (B) the astrocyte niche, and the (C) neuron niche.

### 5.2. The Vascular Niche Supports Residual Invasion

Glioma cells may recapitulate migration patterns that take place during CNS development. Organotypic brain slices co-cultured with patient-derived GBM cells revealed that glioma cells take a unipolar stretched morphology with leading invasive protrusions like glial progenitor cells along blood vessels [95]. These structures are also termed focal adhesions or invadopodia and are essential to the migration of cells [99,100]. These invadopodia can sense ECM, degrade it, and induce cytoskeleton remodeling to move the cell body forward and infiltrate the surrounding parenchyma [101]. Intracellular calcium facilitates these cytoskeletal changes. Additionally, increased intracellular calcium concentration can activate migratory programs in glioma cells potentially in collaboration with the transient receptor potential vanilloid type 4 (TRPV4), which is a Ca^2+^-permeable non-selective cation channel that belongs to the transient receptor potential vanilloid superfamily [102]. A recent study demonstrated that TRPV4 is present at focal adhesions and is involved in the invasion and growth of subcutaneously and intracranially-transplanted GBM in mice [103]. Moreover, this receptor is highly expressed preferentially in high-grade gliomas over low grades and normal brains [103]. Cell division control protein (Cdc) 42 is a Rho-GTPase that regulates actin-cytoskeleton rearrangements and subsequently cell polarity and migration by specifying the localization of such filopodia [104]. Okura et al. demonstrated, using the TCGA dataset, that Cdc42 expression correlated to worse progression-free survival in the proneural and neural subtypes [105]. Brain slices co-cultured with GBM cells have revealed Cdc 42-dependent interaction between GBM and pericytes in co-opted vessels [106]. Next, GBM binds endothelial cells lining the blood vessels attracted by chemoattracts secreted by endothelial cells.

The interaction between GBM cells and endothelial cells in blood vessels is achieved through a combination of ECM and cell surface receptors. GBM cells bind to the endothelial ephrin-B2, as evidenced through intravital imaging of murine GBM and patient-derived cell culture models [107]. Additionally, GBM cells expressing bradykinin receptor 2 (BKR2) bind to bradykinin, which is physiologically present in the brain’s blood vessels and is an endothelial cell product that is increased during tumor progression [108,109]. It is not a gradient of nutrients or oxygen that attract GBM cells to blood vessels since functional blood vessels are not present in these studies using brain slice co-cultures with GBM, but rather chemoattractants secreted by parenchymal cells.

Distinctive paracrine signals can direct perivascular invasion. Wnt stimulates single-cell invasion onto blood vessels in a zebrafish glioma model [110]. Additionally, Wnt7 promotes single-cell perivascular invasion into brain slices of Olig2 + GBM cells, whereas Olig2-cells migrate over blood vessels as clusters of cells [83]. Chemokine signaling is an essential driver of perivascular migration into the brain parenchyma. IL8 produced by endothelial cells promotes glioma spheroids invasion in a three-dimensional collagen matrix; it stimulates invasion and tumor growth in vivo in a cell line-derived orthotopic model [111]. IL8 is necessary for tumor progression, as it can regulate stem cell marker expression in GBM and other cancers [112]. Furthermore, downregulating IL8 by using targeted siRNA and neutralizing antibodies inhibited invasion in Matrigel of several glioma cell lines [113]. Strikingly, IL8 activates the invasion of glioma stem cells precisely [114]. The invasion is possibly mediated through the IL8 activation of NFkB [113]. Additionally, bradykinin binding to BKR2 both increases intracellular calcium to support cytoskeletal remodeling as well as activates the NFkB pathway to promote migration of GBM cells [58,109]. This suggests that specialized GBM can migrate across blood vessels in response to chemoattractants secreted by endothelial cells. Surprisingly, a neurotransmitter may play a role in vessel co-option, since one of the most abundant glutamate receptor subunits, the α-Amino-3-hydroxy-5-methylisoxazole-4-propionic acid (AMPA) subunit GluR1, associates with β1- integrin at focal adhesions and correlates with sticking to the ECM component collagen, which is most abundant in endothelial basement membranes and brain meninges [115]. Interestingly, the high intracellular calcium present in these invading GBM cells may facilitate glutamate exocytosis, an essential function in GBM, as described in the next section.

### 5.3. The Neuron Niche Supports Growth

Glioma cells rely on neurochemical interactions with neurons for survival and growth. In contrast to low grades, higher grades of glioma are highly interconnected with the brain parenchyma, which may also be a reflection of the high invasion capacity of advanced gliomas [116,117,118]. Several studies have found that glioma cells form synapses with neurons in human glioma samples, and these are more common in high grades than low grades [116,117,118,119]. Venkeratamani and colleagues observed that such synapses produce synaptic currents mediated by the AMPA subtype’s glutamate receptors [119]. Depolarization of glioma membranes promoted proliferation, whereas blocking electrochemical signaling inhibited glioma xenografts’ growth. Intriguingly, a second study found synapses between breast cancer cells and glutamatergic neurons involving activation by glutamate ligands of N-methyl-D-aspartate receptors (NMDARs) [120]. 

High-grade gliomas release excitotoxic glutamate concentrations [119,121]. Glutamate is a neurotransmitter present in most of the excitatory synapses. It typically stimulates neuronal and oligodendrocyte migration and enables glioma cells’ migration [119,121,122]. Moreover, it may promote glioma cells’ viability, stimulating tumor growth [123,124,125]. Glutamate receptors show a higher expression in GSC [126]. One of the prominent roles of glutamate is maintaining autophagic homeostasis by regulating mechanistic Target of Rapamycin (mTOR) signaling [127]. Autophagic flux is essential for the viability of GSC. It may enable chemo-resistance [127,128,129]. Wang et al. demonstrated that Notch1 + glioma cells bound with its Jagged1 ligand on axons [130]. This interaction led to a positive feedback loop governed by SOX2 and 9 on glioma cells, which decreased Notch1 promoter methylation, leading to its transcription to reinforce glioma–neuron association [130]. This indicates an innovative form of direct and biologically relevant synaptic communication between neurons and tumor cells. Astrocytes regulate glutamate homeostasis in the brain, with potential consequences for migration is the astrocyte [131]. 

### 5.4. The Astrocyte Niche Promotes Invasion and Mesenchymal Transformation 

Tumors induce and surround themselves with reactive astrocytes. Reactive astrocytes are generated by neoplasms, as exemplified in the GL261 murine glioma model by increased glial fibrillary protein (GFAP) + cells [74]. Reactivation of perivascular astrocytes is induced during mouse aging marked by increased GFAP coupled with a loss of blood vessel integrity characterized by loss of AQP4 localization [132,133]. Moreover, loss of AQP4 and vascular integrity has been observed in human glioma tissues [134,135,136]. Strikingly, GBM cells were observed to migrate along blood vessels near AQP4+ astrocytes and induce loss of AQP4 in astrocytes as the tumor grows [84]. Since tumor cells induce astrocyte reactivation, this already suggests their essential role in tumor progression. 

Astrocytes promote GBM invasion and growth of tumors in a paracrine fashion (Figure 1, Box 2C) [137,138,139]. The U251N glioma cell line showed more substantial transmigration through a Boyden invasion chamber when co-cultured with astrocytes due to secreted MMP2 [137]. Murine reactive astrocytes produce TGF-β in the patient-derived preclinical glioma model [140]. Tumor-secreted receptor activator of NFkB ligand (RANKL) activates NFkB in astrocytes leading to their producing transforming growth factor β (TGF-β), which stimulates invasion in vivo in xenografts of the glioma cell line U87 [141]. A third essential cytokine produced by reactive astrocytes is the alarmin IL33 [74]. IL33 is part of the IL1 family and binds the ST2 receptor, and it can promote growth and dissemination by remodeling the tumor microenvironment [142]. IL33 has been found to activate NFkB-transcription of MMP2 and 9 initiating migration and invasion of glioma cell lines U251 and U87 as assayed by using transwell chambers [143,144,145]. Moreover, in autocrine-fashion, it promoted transcription of tenascin C, a multimeric ECM glycoprotein that is downregulated in the adult brain except for canonic neurogenic zones in the hypothalamus; however, it is also expressed by reactive astrocytes [146,147]. Accordingly, gliomas and mostly GBM highly express tenascin C compared to the normal brain, and it is correlated with poor prognosis [145,148]. Tenascin C binds the TLR4 to induce activation of tumor-associated macrophages and microglia and patterning towards an M2 phenotype essential for tumor growth and progression [146,149]. It also activates NFkB-dependent transcription of mesenchymal genes leading to a mesenchymal switch in tumor cells [150]. Taking all this together, the intracellular invasion pathways converge on NFkB, underscoring the essential role this transcription factor plays in glioma invasion.

Concludingly, NFkB has a two-fold role: when activated in cancer cells, it supports migration out of the primary tumor and chemo-resistance, and, when activated in the reactive astrocytes, it reorganizes the peritumoral niche to favor invasion and survival of disseminated glioma cells [151]. Since NFkB also controls transcription of the SASP, some of these components could substitute for all the factors mentioned in the above sections [152]. Part of the non-cellular elements that drive invasion and growth into the three niches can also be produced and secreted by senescent cells.

## 6. Factors Secreted by Senescent Cells Create a Supportive Niche for GBM Recurrence

There are two arguments why cellular senescence could contribute to GBM. First, aged brains from healthy individuals over the age of 60 are abundant in senescent astrocytes, and GBM is mostly diagnosed in this age group (Figure 2) [28]. Second, DNA-damaging anti-cancer therapies have been shown to induce senescence in healthy cells. This includes the first-line treatments for GBM, such as radiotherapy and Temozolomide. Indeed, ionizing radiation induces senescence in the mouse brain, which has been suggested to be a cause for functional irregularities [32,153]. Additionally, Temozolomide causes senescence in astrocytes in vitro [38,154]. The next step is to verify if senescent cells are present in GBM samples, most notably in the peritumoral zone. This area receives both radiation and chemotherapy since this tumor niche is critical for tumor recurrence.

Senescent brain cells mass-produce and secrete factors ranging from ECM components and remodelers to chemoattractants, some of which can promote tumor viability and invasion into any of the three niches (Table 1). For example, astrocytes from aging primate brains synthesize excess hyaluronan, supporting both the single-cell invasion and NFkB activation [155]. Senescent astrocytes also produce excess fibronectin, potentially promoting cell survival and collective invasion [156]. Senescent astrocytes have both MMP 2 and 9, which are essential factors for GBM invasion into multiple brain niches [156,157]. Typically, astrocytes are required to clear neurochemicals from the synaptic cleft to avoid damaging neurons [131]. Senescent astrocytes are less well equipped to do this because of decreased expression of glutamate transporter GLT1, leading to excess synaptic glutamate and long-term potentiation of neurons coupled with excitotoxicity [158,159]. This could be exploited by cancer cells to amplify the local levels of glutamate to support growth and invasion. Senescent astrocytes and endothelial cells are also known to secrete a plethora of cytokines and chemokines, among which IL6 and IL8 can activate NFkB and STAT3 signaling in glioma cells to maintain their stemness and promote their invasiveness [156]. Conclusively, conditioned media from in vitro-induced senescent astrocytes promoted the proliferation of glioma cell lines by increasing c-Myc [160]. More importantly, this leads to improved survival from TMZ [160]. This initial indication of the direct role of the SASP in therapy resistance prompts further studies, especially in vivo studies, to understand whether anti-senescence therapies targeted at niche cells could be applied to glioma. 

## 7. Conclusions: Senescence in the Tumor Niche Is Likely to Favor Advanced GBM and Post-Treatment Recurrence

Invading cells create new supportive niches to forward recurrence [161]. Compiling up-to-date literature, we here define three distinct niches and their roles in sustaining residual disease. There are several arguments as to why cellular senescence is likely to be a driver of advanced GBM and a culprit in residual disease:

(A) GBM is more commonly diagnosed in the elderly, with a median age at diagnosis of 62. At this age, there is a considerable elevation of senescent cells in tissues, including the brain. (B) The combination of TMZ with radiotherapy that is administered after surgery induces cellular senescence in the peritumoral niche. This is where residual GBM cells reside. (C) Although the residual niches are very different entities, they secrete factors that are also part of the SASP, which sustain residual glioma through therapy and drive therapy resistance and recurrence. The SASP correlates with every step of cancer progression, and the SASP factors can drive tumor growth, invasion, and therapy resistance. Altogether, we, therefore, conclude that cellular senescence is likely to cause residual GBM, and the development of tailored anti-senescence therapies should be investigated as a new concept in the treatment of advanced glioma. 

## 8. Future Perspectives: Strategies to Target Senescence in Glioma

Several anti-senescence compounds are under advanced development, with many options being evaluated in at least a dozen start-ups and scale-ups [162]. The design of anti-senescence compounds is focused on key drivers of cellular senescence. At the core of the senescence-associated growth-arrest lies p53 activated by the chronic unrepaired DNA damage. Such insults typically induce apoptosis; however, senescent cells present elevated expression of anti-apoptotic proteins, such as (B-cell lymphoma) BCL that counteract the apoptotic function of p53 [163]. In a recent study, Baar [30] identified a pivoting factor in the viability of senescent cells, the Forkhead box protein O (FOXO) 4 can sequester p53 to favor growth arrest and prevent it from triggering apoptosis. Accordingly, dissociating p53 from FOXO4 using a cell-penetrating blocking peptide leads to p53 relocalization to the mitochondria, thereby causing caspase-dependent apoptosis. Whether FOXO4-p53 interference would work against GBM remains to be investigated. 

Several alternative anti-senescence compounds are also focused either on p53 reactivation or on blocking anti-apoptotic signals. The latter is exemplified by BCL blockers [162]. The former is achieved either through peptides inhibiting FOXO4-p53 or with Mouse Double Minute (MDM2) blockers that prevent degradation of p53 [30,162]. It is essential to consider the application of these compounds to GBM for maximizing treatment. A different proposed strategy is by applying anti-senescent drugs in a two-strike model: first induction of cancer cell senescence by classical chemotherapy followed by anti-senescence treatment. For this strategy to be considered a candidate treatment for GBM, it needs to overcome two obstacles. Firstly, GBM shows alterations in the main senescence drivers. As is common in other neoplasms, gliomas arise from a multistep process driven by genetic alterations, including loss of tumor-suppressor genes, e.g., TP53, cyclin-dependent kinase inhibitor 2A and 2B (CDKN2A/B) and PTEN 10 [1]. Whether these mutations influence GBM cell’s potential for entering into the senescence-associated growth arrest is unclear and begets further investigation. Secondly, the highly plastic nature of GBM could lead to senescence-evasion. The senescence dogma stipulates a permanent-growth arrest; however, recent evidence suggests that this growth arrest is less robust than once thought and can be reversed significantly in cancer cells, enhancing stemness and leading to recurrence [164].

Research into senescence in glioma is at its infancy, with the first indication that the two-strike strategy might work for GBM coming from a recent in vitro study [165]. A high one-time dose of radiation was sufficient to induce senescence in glioma cell lines, which are targeted by a BCL-blockade [165]. Since one-dose radiotherapy is likely to be unsuitable for clinical application, numerous groups are investigating alternative senescence-inducers. Recently-discovered alternative glioma cell senescence-inducers include MAPK-activated protein kinase 2 inhibitors, the plant sugar flagellin, and the sugar D-galactose [160,166]. The next step is to apply anti-senescence therapies against these kinds of cancer cell senescence; however, it is also essential to investigate whether the standard-of-care is sufficient to induce targetable senescent not only glioma but also niche cells. A prolonged combined treatment regimen of p53-modulating anti-senescence compounds, like the ones mentioned previously, following chemo- and radio- therapy may lead to a new treatment concept.

Whether anti-senescence treatments would be efficient in GBM still needs to be investigated. At present, most research focuses on identifying ways to target senescent glioma cells; however, since this review unveiled a potential role of senescent niche cells in maintaining residual disease, it warrants investigation into targeting the niche. Such a treatment strategy can implement anti-senescence therapies in cancer care, focusing on targeting residual disease to prevent a recurrence. At the dawn of anti-senescence treatments, it is time to understand the different types of senescence and their roles in residual disease to develop tailored anti-senescence treatments to counter chemo- and radio- therapy resistance to improve patient survival. 

## Figures and Tables

**Figure 1 cancers-13-01560-f001:**
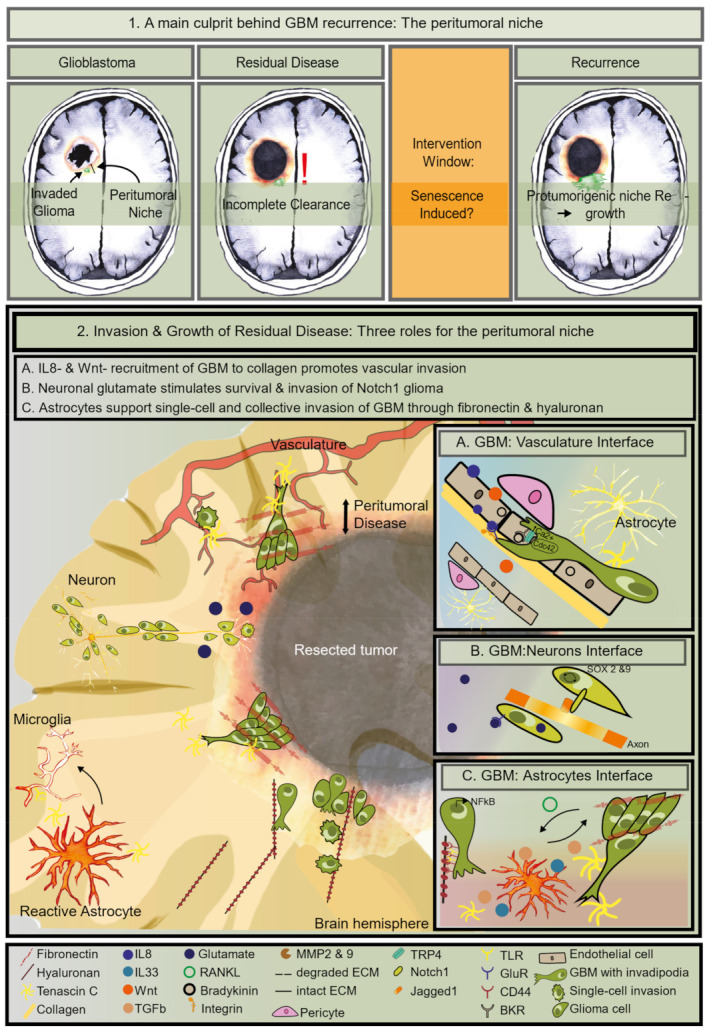
Clinical and biological definition of residual disease; (1) Illustration of different stages during patient-treatment arguing for an intervention window following standard-of-care. Treatment-induced senescence may be a target. (2) Illustration of residual disease. Graphical representation of three situations of pro-tumorigenic brain niches which can both stimulate invasion as well as promote survival of cancer cells. The left side depicts the cellular-components in the three niches and the right side shows the molecules and receptors responsible for the bi-directional communication between glioma cells and their niche. (**A**) IL8- & Wnt-induced invasion over blood vessels is sustained by calcium-mobilization and glioma cell cytoskeletal reorganization to move over collagen in the basement membrane. (**B**) Glutamate-stimulated invasion of Notch1-positive glioma cells to its cognate receptor on axons. This interaction activates SOX2 & 9 signaling in glioma cells; (**C**) bidirectional communication between glioma and astrocytes. NFkB is active in these glioma cells. Glioma RANKL- induced TGF-b, IL33, and MPPs secreted by reactive astrocytes are necessary for glioma invasion both as single cells and as a strand. Hyaluronan induces single cell migration through CD44 and TLR on glioma cells. Fibronectin enables collective invasion over tenascin C fibers.

**Figure 2 cancers-13-01560-f002:**
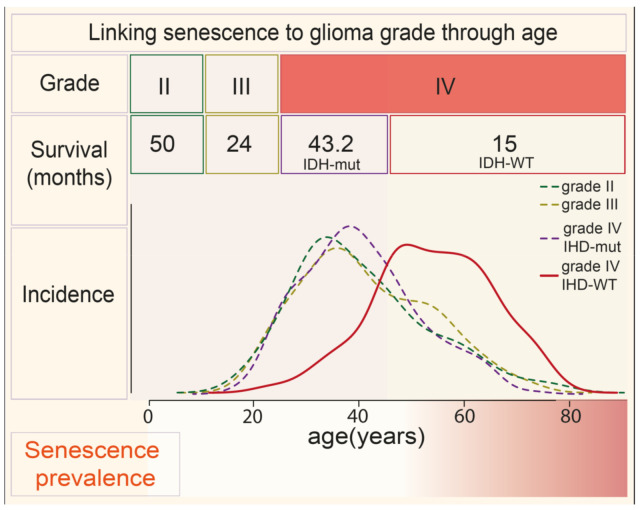
Linking glioma grade to age and senescence. The table depicts median survival and graphs incidence per glioma grade [2,27]. Stage II (dark green dotted line), stage III (light green dotted line), and GBM with IDH mutations (IDH-mut, purple dotted line) occur in younger people. Grade IV, GBM without IDH mutations (IDH-WT, red line) has the worst prognosis and mostly occurs in the elderly with a median around 60 years old. The bottom row compares glioma incidence to the age of senescence onset and senescence prevalence [28].

**Table 1 cancers-13-01560-t001:** Selected SASP components and their potential role in GBM, a residual disease.

Role in Glioma Progression	Type of Component	Name of the SASP Component	Cell Type
Invasion	ECM	Fibronectin	Astrocyte
Invasion	ECM	Hyaluronan	Astrocyte
Invasion	ECM	MMP2	Astrocyte
Invasion	ECM	MMP9	Astrocyte
Invasion	Ion balance	Calcium	
Invasion	Cytokine	IL6	Astrocyte, endothelial cell
Invasion	Cytokine	IL8	Astrocyte, endothelial cell
Glioma cell proliferation	Nutrients	Glutamate	Astrocyte
GSC maintenance	Cytokine	IL33	Astrocyte

Data for the table was selected from [156].

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
