# Peer review of "Residual Disease in Glioma Recurrence: A Dangerous Liaison with Senescence"

_cancers, 2021, doi:10.3390/cancers13071560_

Round 1
Reviewer 1 Report
This is a comprehensive, up-to date review on the glioblastoma multiforme (GBM), focused on the sub-population of therapy resistant, invasive, tumor cells, responsible for tumor relapse. These cells have characteristics of cancer stem cells and their phenotype is largely regulated by their microenvironment. Senescent cells, which accumulate because of the age or in response to therapy, may be an important component of the microenvironment and enhance invasiveness stemness and therapy resistance.
I have no major concern about this review, but I suggest an accurate revision of the text.
Listed below are a few examples of miswriting.
Lane 15 approached instead of approaches
Lane 18 ….. a chronic secretory phenotype that are…….
Lane 111 …… senescence is induced……. but they are
Lane 183 Myelocytomatosis (Myc) should be Myelocytomatosis oncogene
In several instances GBM is written Gbm
etc
Author Response
Response to reviewer 1 comments
[Authors] We greatly thank the reviewer for the correct summary and the encouraging words. We have rewritten a large part of the manuscript and included your suggestions.
Reviewer 2 Report
The authors have reviewed aspects of glioma recurrence and its relationship to senescence.
The main concern with the article is that the two broad sections (brain niches and senescence) are not connected in any meaningful way. What is the relationship between niches and senescence? Are senescent cells found in particular niches? The author's must integrate the different aspects of the review more effectively.
The article contains numerous editing mistakes which should be addressed. For example Gbm is used many times instead of GBM. The formatting in Table 1 is messy.
The term "multiforme" is no longer used.
Author Response
Response to reviewer 2 comments
“The main concern with the article is that the two broad sections (brain niches and senescence) are not connected in any meaningful way. What is the relationship between niches and senescence? Are senescent cells found in particular niches? The author's must integrate the different aspects of the review more effectively.”
[Authors] We understand the concern of the reviewer and agree this is very important to clarify. In brief, yes, senescence is elevated in the tumor niches with age and can be induced by the standard-of-care of RT and TMZ (not TTFields). Factors secreted by senescent cells were shown to drive advanced GBM and promote recurrence. We now stressed the link between the two main sections of this review: senescence and the glioma niches in the 2nd and 6th sections.
“The formatting in Table 1 is messy”
[Authors] As per the suggestion of the reviewer, we decided to change the table into a figure. The main point here was to indicate that advanced GBM occurs mainly at older age and this is also when there is an elevation in senescent cells.
“The term "multiforme" is no longer used”
[Authors] We have removed “multiforme” as per correct indication of the reviewer.
Round 2
Reviewer 2 Report
I am happy to accept this manuscript for publication.Author Response
We thank you for your comments.